# Epidemiology of Hemorrhagic Fever with Renal Syndrome and Host Surveillance in Zhejiang Province, China, 1990–2021

**DOI:** 10.3390/v16010145

**Published:** 2024-01-19

**Authors:** Fan Su, Ying Liu, Feng Ling, Rong Zhang, Zhen Wang, Jimin Sun

**Affiliations:** 1Health Science Center, Ningbo University, Ningbo 315211, China; sufan412@163.com; 2Key Lab of Vaccine, Prevention and Control of Infectious Disease of Zhejiang Province, Zhejiang Provincial Center for Disease Control and Prevention, Hangzhou 310051, Chinarzhang@cdc.zj.cn (R.Z.)

**Keywords:** hemorrhagic fever with renal syndrome, epidemiological characteristics, rodent

## Abstract

Hemorrhagic fever with renal syndrome (HFRS) is caused by hantaviruses (HVs) and is endemic in Zhejiang Province, China. In this study, we aimed to explore the changing epidemiology of HFRS cases and the dynamics of hantavirus hosts in Zhejiang Province. Joinpoint regression was used to analyze long-term trends in the incidence of HFRS. The comparison of animal density at different stages was conducted using the Mann–Whitney Test. A comparison of HV carriage rates between stages and species was performed using the chi-square test. The incidence of HFRS shows a continuous downward trend. Cases are widely distributed in all counties of Zhejiang Province except Shengsi County. There was a high incidence belt from west to east, with low incidence in the south and north. The HFRS epidemic showed two seasonal peaks in Zhejiang Province, which were winter and summer. It showed a marked increase in the age of the incidence population. A total of 23,073 minibeasts from 21 species were captured. Positive results were detected in the lung tissues of 14 rodent species and 1 shrew species. A total of 80% of the positive results were from striped field mice and brown rats. No difference in HV carriage rates between striped field mice and brown rats was observed (*χ*^2^ = 0.258, *p* = 0.611).

## 1. Introduction

Hemorrhagic fever with renal syndrome (HFRS), a rodent-borne disease caused by hantaviruses (HVs) of the family Bunyaviridae, is characterized by fever, hemorrhage, kidney damage, and hypotension [1]. According to previous studies, over 90% of all cases of HFRS in the world have occurred in China in recent decades [2]. Despite the recent reduction in human morbidity and mortality from HFRS as a result of improved prevention and treatment strategies, HFRS remains a significant public health challenge in China. To date, China remains the most endemic country for HFRS [3]. Between 2004 and 2019, a total of 209,209 cases of the disease were registered, of which 1855 died [4]. Zhejiang Province is an endemic area for HFRS, with the first case reported in 1963; since then, the epidemic continued to rise, and there were several outbreaks of HFRS in the 1970s and 1980s. In the 1990s, the epidemic was effectively controlled through rodent eradication and vaccination [5]. However, the spatial distribution of HFRS is expanding and new sources of the disease are emerging in eastern Zhejiang Province. Every Old World HV genotype that is now known to exist can cause HFRS, including the Dobrava/Belgrade virus (DOBV/BGDV), Puumala virus (PUUV), Seoul virus (SEOV), and Hantaan virus (HTNV) [6,7,8]. Two HV genotypes, HTNV and SEOV, primarily carried by the rodents striped field mice (*Apodemus agrarius)* and brown rats (*Rattus norvegicus)*, respectively, are responsible for the majority of cases in China [9]. Most HFRS patients are infected through direct contact with aerosolized feces or body fluids from infected rodents [10,11]. China began to establish nationwide surveillance sites for human and inter-animal outbreaks in the 1980s. Host animal monitoring focuses on relative animal density, population, and HV carriage. Some studies have shown that the density and carriage rate of host animals directly affect the incidence of HFRS. Tian H et al. analyzed the relationship between HFRS outbreaks and rodent density in Xi’an City from 2005 to 2012 and found that monthly HFRS outbreaks in Xi’an City were significantly correlated with the density of rodents 2 months earlier [12]. Bai Y et al. found that the HFRS outbreaks in Chongqing Municipality of China were significantly correlated with the density of rodents, temperature, and rainfall [13]. Jiang F et al. analyzed the relationship among HFRS outbreaks, host animal density and carrier rate, and meteorological factors in Qingdao City from 2007 to 2015, and they found that temperature and humidity could affect the reproduction and carrier rate of host animals, which in turn could affect HFRS outbreaks [14]. Moreover, new species are constantly being found to carry HV that are geographically specific [15,16,17]. The new HV transmitted by Ussuri white-toothed shrews (*Crocidura lasiura*) and Asian house shrews (*Suncus murinus*) was discovered in Zhejiang Province in 2014 [18].

In addition to national surveillance sites (Tiantai County), Zhejiang Province has been implementing animal monitoring in 15 other counties throughout the province since the 1980s. More than 40 years of host animal monitoring data have been collected here, but they have not been thoroughly reviewed. Given the data availability, our study examined the changes in host animal density and HV carriage rate in Zhejiang Province from 1990 to 2021. More importantly, we analyzed the changing dynamics of animal populations as well as the prevalence of HV in different populations. The objective of this study was to analyze the epidemiological features of HFRS in Zhejiang Province during a 32-year period (1990–2021) and review the temporal dynamics of the HFRS host animal populations, relative density, and their HV infection status.

## 2. Materials and Methods

### 2.1. Study Sites

Zhejiang Province is located at 27°–32° N and 118°–123° E. It is bounded on the east by the East China Sea, on the south by Fujian Province, on the west by Anhui Province and Jiangxi Province, and on the north by Shanghai and Jiangsu Province. The plains of northern Zhejiang, the middle hills of western Zhejiang, the hills of eastern Zhejiang, the Jinqu Basin in central Zhejiang, the mountains of southern Zhejiang, the plains of the southeastern coast, and the coastal islands are the province’s six terrain areas (Figure 1). Mountains and hills account for 74.6% of land in Zhejiang Province, flat ground for 20.3%, and water for 5.1%.

### 2.2. Data Collection

According to the health industry standard of the People’s Republic of China for the diagnostic criteria of HFRS, HFRS cases were classified as suspected cases, clinically diagnosed cases, and confirmed cases. (1) A patient with an epidemiological history and clinical manifestations with fever or gastrointestinal symptoms was defined as a suspected case; (2) a suspected case with hypotension, renal function impairment, increased peripheral blood cell counts and thrombocytopenia, and a positive urine protein was defined as a clinically diagnosed case; (3) a clinically diagnosed case with one or more of the following criteria was defined as a confirmed case: the serum-specific IgM antibody is positive; the specific IgG antibody is 4 times higher than that in the acute phase; Hantavirus RNA is positive or Hantavirus is isolated.

Data on HFRS cases were obtained from two sources. First, data from 1990 to 2004 including the number of cases per month at the county level and demographics (age, sex, and occupation) of patients with HFRS were obtained from the archives of the Zhejiang Provincial Center for Disease Control and Prevention (CDC). Second, data from 2005 to 2020 were obtained from the China Reportable Disease Reporting System and included case information, including residential address, time of onset, time of diagnosis, time of death, sex, age, outcome, and occupation.

### 2.3. Host Animal Surveillance

Data on rodent hosts within the HFRS endemic region have been available since 1990 in Zhejiang Province through monitoring by Zhejiang CDC. The night trapping method was used to capture hosts with peanuts as bait. More than 200 traps per patch (about 7850 m^2^) were placed in the field for three consecutive nights. The wild and residential areas were both taken into account when placing the traps. The trapped hosts’ species were identified by morphology and their lungs were collected to test for the HV antigen with monoclonal antibodies by direct immunofluorescence assay. We used relative density to evaluate the abundance of host animals. Relative density = (number of hosts captured/number of traps) × 100%.

### 2.4. Statistical Analysis

We examined the trend in incidence from 1990 to 2021 using joinpoint regression and described this trend in terms of annual percentage change (APC). Student’s *t*-test was applied to assess if the APC significantly differed from 0. All joinpoint regression analyses was conducted with the Joinpoint software (Version 4.9.1.0). The epidemiological characteristics of HFRS cases, including geographic distribution, seasonal pattern, gender, age, and occupation, were analyzed using descriptive methods. We used the Mann–Whitney test to compare the density levels of host animals at different periods of time. The chi-square test was used to compare differences in the rates of HV carriage in animals over time and positive detection rates by species. The Mann–Whitney test and chi-square test were completed by R software (Version 4.2.1). The difference was considered statistically significant when *p*-value < 0.05.

## 3. Results

### 3.1. Epidemiological Feature

A total of 51,370 cases and 240 deaths were reported in Zhejiang Province during 1990–2021. The annualized average incidence ranged from 0.3532 to 13.1077/100,000 persons. Over 1000 instances were recorded annually from 1990 to 2002. More than 80.00% of cases were reported during 1990–2004. In 2021, fewer than 200 cases were reported. There were two joinpoints through the final regression models. The annual incidence sharply decreased from 1990 to 2001, with an APC of −10.93% (*t* = −16.9, *p* < 0.001). From 2001 to 2004, the annual incidence continued to decrease, with an APC of −21.99% (*t* = −1.5, *p* = 0.153). This decrease slowed down from 2004 to 2021, with an APC of −7.27% (*t* = −7.8, *p* < 0.001). By 2014, the annual incidence had decreased to less than 1/100,000 (Figure 2).

#### 3.1.1. Regional Distribution of HFRS

Overall, there was a high incidence belt from west to east, with low incidence in the south and north. Since 2006, the range of the high-incidence area in the west has shrunk significantly. Few cases have been reported in the island areas, and no cases have been reported in Shengsi County. In 1990, 68 counties reported cases of HFRS, by 2021, the number of counties with case reports reached 89. Kaihua County had the highest cumulative incidence rate (758.8695/100,000) in Zhejiang Province, followed by Xinchang County (689.3358/100,000). Dongtou, Haiyan, Taishun, Putuo, Dinghai, and Wencheng counties had less than 10 cumulative cases each, and Dongtou County did not report a case until 2011. A total of 77.49% of the cases came from rural areas and the incidence rate in rural areas (138.9354/100,000) was higher than that in urban areas (72.1010/100,000) (Figure 3).

#### 3.1.2. Seasonal Distribution of HFRS

The HFRS epidemic showed two seasonal peaks in Zhejiang Province, which were winter (November to January) and summer (May to June). Overall, the incidence rate in winter was higher than that in summer, except for the years 1999, 2013, 2014, and 2020, when more cases were reported in summer (Figure 4). Among the high prevalence counties of HFRS in Zhejiang Province, the seasonal distribution manifested itself in three types. Xiangshan, Jiande, and Xinchang only showed a winter peak, while Yinzhou and Zhenhai had most cases in summer. Other counties showed two seasonal peaks simultaneously (Figure 5). September had the lowest incidence rate of the year.

#### 3.1.3. Population Distribution of HFRS

Cases were found in all age groups, with 81.28% of the cases occurring in the 20–59 years age group in Zhejiang Province. And the median age was 47 years (IQR: 37–57). Between 2005 and 2021, the median age of the cases increased from 44 to 52 years old (Figure 6). In 1991, the proportion of individuals aged 60 and up was 8.60%, and this proportion continued to climb until 2020, reaching 34.80%. Overall, farmers accounted for 76.94% of the cases, with factory workers accounting for 9.35%. Of those aged 20–59, 67.83% were farmers and 15.75% were factory workers. Furthermore, the proportion of domestic workers kept on rising. Among all cases, the overall male-to-female ratio was 2.56:1.

### 3.2. Host Animal Surveillance

A total of 16 counties in Zhejiang province undertook host animal monitoring from 1990 to 2021, capturing a total of 23,073 small animals. For the monitoring of relative density, a total of 367,565 effective clips were deployed, 19,132 animals were captured, and the overall relative density was 5.20%. There was a significant change in relative density between 1990 and 2021, with a maximum of 10.50% in 1992 and a minimum of 3.03% in 2010. The relative density from 2005 to 2021 was lower than that from 1990 to 2004 by the Mann–Whitney test (*p* = 0.021). There were 803 positive results among the 19,076 lung tissue samples examined. The HV carriage rate ranged from 1.15% (2017) to 15.71% (2003) (Table 1).

#### 3.2.1. Host Animal Populations

By morphology, 17 species of rodents and 3 species of shrews and weasels were successfully identified, and an average of 11 species of animals were captured annually. Animals captured in the wild are striped field mice, confucian niviventer (niviventer *confucianus)*, reed vole (*Microtus fortis)*, Himalayan field rats (*Rattus nitidus)*, and chestnut white-bellied rats (*Niviventer fulvescens)*. In residential areas, brown rats, house mice (*Mus musculus)*, losea voles (*Rattus losea*), oriental house rats (*Rattus flavipectus)*, and Asian house shrews are primarily captured. Striped field mice, brown rats, oriental house rats, house mice, and losea voles were captured every year. The dominant species were striped field mice (45.38%) and brown rats (23.46%). The population of brown rats surpassed that of striped field mice in 1990, 1991, 1996, and 2001. However, since 2009, the number of brown rats has drastically decreased. The first chestnut white-bellied rat activity was detected in the wild in 2005, and they have been captured every year since. An emerging rodent, the Bower’s white-toothed rat (*Berylmys bowersi)*, was observed in 2021. A total of 73.49% of the shrews were Asian house shrews, and their activity was not limited to 1998 and 2005 (Table 2).

#### 3.2.2. HV Carriage Rate

HV antigen positivity was found in the lung tissues of 12 rodent species and 2 shrew species, with striped field mice and brown rats accounting for more than 80% of the total. Up to 10 animals tested positive for the HV antigen in a year (2006). Every year, positive samples of striped field mice were found, with an HV carriage rate of 6.39%. Brown rats had just two years of negative results, with an HV carriage rate of 6.13%. There was no statistical difference in the HV carriage rate between striped field mice and brown rats by the chi-square test (*χ*^2^ = 0.258, *p* = 0.611). Aside from the two rodent species mentioned above, oriental house rats, house mice, losea voles, and confucian niviventer had positive detections throughout the year. No evidence of HV infection in losea voles was reported from 2009 to 2021 (Figure 7). The two species of HV-carrying shrews were the Asian house shrew and the Asian gray white-toothed shrew (*Crocidura attenuate)*.

## 4. Discussion

The incidence of HFRS in Zhejiang Province showed a continuous decline from 1990 to 2021, which was in accordance with the overall epidemiological trend in China [19]. Prior to the 1990s, HFRS first became widespread in the northern regions of inland China, gradually spreading throughout the country, while Zhejiang Province reached its highest ever incidence of HFRS in 1986. In order to control the rodent-borne disease epidemic, the Chinese government implemented a nationwide rodent control campaign in 1986, which has been in place since then. In addition, the Zhejiang Province government provided free HFRS vaccination services to residents in high-risk areas from 1995 to 2011. Rodent control and vaccination have effectively controlled the HFRS epidemic. During the COVID-19 pandemic, the incidence rate in Zhejiang Province dropped to 0.49 per 100,000 population in 2020, and the preventive and control measures for the COVID-19 pandemic also contributed to a sudden decrease in the incidence of various zoonotic diseases [20]. At the same time that the incidence rate kept decreasing, the epidemiology of HFRS also showed a marked increase in the age of the incidence population. The composition ratio of people aged 60 and over in the 2020 morbidity population is 34.80%, which is four times higher than in 1990. Between 2005 and 2021, the median age of incidence rose by 8 years. The ageing of the incidence population was much higher than the population-wide level. The age composition of the farming community has changed significantly as a result of rapid economic development. Currently, young people are more inclined to work in cities, and those who work in agriculture in rural areas are mainly middle-aged and older people. Farmers are at high risk for HFRS, leading to an overall aging of the population.

The high-incidence provinces in southern China are the Hunan, Jiangxi, Zhejiang, and Fujian provinces [7,21,22,23], which are four neighboring provinces located in the latitudinal belt from the Tropic of Cancer to the 30°N latitude, from west to east. The high-prevalence areas within Zhejiang Province also showed an expansion from west to east in the same latitudinal belt [5], while the incidence of HFRS in the south and north is significantly lower than in this region. The topography of Zhejiang Province from west to east shows mountains, basins, and hills, while the north and south are all plains. This shows that regions with more mountains are more prone to the prevalence of HFRS. Among the five counties with the highest prevalence rates, Kaihua County was located in the westernmost part of Zhejiang Province, and Shengzhou County, Xinchang County, and Tiantai County formed the east-central clustering area [24]. Longquan County, which is relatively far from the clustering area, is adjacent to Nanping City, a high-incidence area in Fujian Province. The range of the western high-incidence region contracted significantly after 2006, which might be attributed to the implementation of the vaccination program in high-risk areas since 1995. The southern and eastern islands were the low-prevalence areas of HFRS, with Shengsi County being the furthest inland of the eastern islands and having no reported cases so far. The incidence rate in areas far inland was consistently low compared with the high-incidence areas, which might be explained by the relatively low population mobility and the limited range of the rodent reservoirs due to the sea barrier.

The 16 host animal monitoring sites were widely distributed in HFRS-endemic areas of Zhejiang Province. Host animal surveillance indicated that striped field mice and brown rats, which were the natural reservoirs of HV, were the predominant small mammals in Zhejiang Province. The striped field mice were commonly found in the wild, except for the confucian niviventer, which has been seen more often in the wild in recent years. Brown rats, oriental house rats, house mice, and losea voles, on the other hand, were active around residential areas. Overall, wild mice were more abundant than house mice, which used to be in excess in the early 1990s. The reduction in the number of house rodents was due to the national rodent control campaign that began in 1986, which was aimed at controlling the prevalence of naturally occurring diseases by reducing the number of nuisance animals in residential areas. Considering the urbanization and aging processes in China, it has been suggested that urbanization has a bidirectional effect on the density of rodents [25,26]. The capture rate of brown house mice decreased in recent years, which might be attributed to the increased impervious surface area in the regions [27]. The previous studies demonstrated that the spatial distribution of HFRS outbreaks corresponded with the range of host animal activity [28,29]. Most HFRS cases in Zhejiang Province were found in rural areas, with a lower number of cases reported in urban areas. In rural areas, there is a risk of disease from house rodents in residential areas, and wild rodents are active in productive areas such as farmland, mountains, forests, and reservoirs. Few wild rodents are active in urban areas, and the number of house rodents is relatively limited.

HV carriage rates are higher than carriage rates for other rodent-borne viruses [30]. And the rates in China are generally higher than in other countries [31]. Despite the declining incidence of HFRS in the human population, the hantavirus infection rate of the host animals did not show a significant decrease. The monitoring results show significant fluctuations in HV carriage rates in animals from 1990 to 2021, with peaks of more than 10.00% in 2002 and 2003. HV carriage rates in Zhejiang Province are relatively high compared to other provinces in China [32]. This suggests that HV is highly prevalent in animals and that there may be some periodicity in the prevalence of HV, influenced by the nature of survival and reproduction of various animals. On the contrary, the relative density of the host animals, which reflected the degree of human exposure to the infectious agents, decreased from the previous years due to national rodent control activities. Lowering the rodent density has contributed to the control of the HFRS prevalence [33,34] but has had a limited effect on the inter-rodent transmission of HV. The high prevalence of HV among animals suggests that Zhejiang Province is still at risk of HFRS outbreaks. In recent years, the prevalence of HFRS in Zhejiang has been highly disseminated. In the current situation of low overall HFRS incidence, new areas are still showing cases, suggesting that low-incidence areas and areas with no reported cases are also characterized by the activity of virulent host animals and the risk of transmission to humans. In view of this, host animal monitoring sites need to be more widely distributed in areas with varying degrees of endemicity.

The HV prevalence rate in both rodent species exceeded 6.00%. The detection rate of the positive HV antigen in the striped field mice was above 15.00% in 2002 and 2003 and over 5.00% during a 10-year period. Tagliapietra et al. suggested that climatic factors, such as temperature and precipitation, may influence the population dynamics and virus transmission of rodents [35]. The spike in the HV carriage rate among rodents may also be partly due to the variation in sampling design and quality, rather than the actual change in virus prevalence [36]. In brown rats, the detection rate of the positive HV antigen was below 10%, peaking at 9.61% in 2020. Both species’ changes in the HV antigen positivity detection rates did not demonstrate significant synchronization. Besides striped field mice and brown rats, various other rodents and insectivorous shrews were also active and detected as antigen-positive for HV. Twenty species with small animal activity were observed in Zhejiang Province, fourteen of which had lung tissue suggestive of HV antigen positivity. Some animals, such as the spiny taiwan niviventer (*Niviventer coninga*), striped dwarf hamster (*Cricetulus barabensis)*, and Asian house shrew, were detected with positive results only three times or less during the 32-year period. Such cases may require re-verification of the authenticity of the test results. Some species that originally tested positive all year round have not been seen carrying HV in the last 10 years, such as the losea vole. The losea vole is the second most dominant rodent species after the brown house mouse in residential areas. Its HV antigen carriage rate was also not synchronized with that of brown rats. Differences in HV prevalence between rodent species need to be analyzed in relation to the HV genotyping carried by each species to determine possible causes. The prevalence of HV among a large number of species is not only found in Zhejiang Province. HV antigen-positive results were found in the lung tissues of small house mice, shrews, and other species in multiple regions [14]. Hong X et al. reported that the incidence risk in urban areas was associated with small house mice [37]. Notably, some emerging species, such as the chestnut white-bellied rat, were frequently detected with antigen-positive results, indicating the need for continuous monitoring of the hantavirus prevalence among animals and the vigilance for the transmission risk in non-dominant hosts.

The prevalence of HFRS is influenced by multiple factors, and recent studies have demonstrated that climatic and meteorological factors affect the occurrence of HFRS [38,39] and that the relationship between HFRS and meteorological factors differs by region [40]. Cao L et al. reported that precipitation was positively correlated with the HFRS incidence rate in a subtropical region and had a lagged effect of 3 months [41], while Rong Z et al. found that the HFRS incidence in Taizhou City, Zhejiang Province, was negatively correlated with the average temperature [42]. All regions of Zhejiang Province are located in the subtropics, and the differences in average temperature and precipitation between regions are not obvious, but there is an abundance of topography. Even under similar meteorological conditions, the activities of host animals within different terrains such as hills, plains, and islands will show different characteristics. How environmental factors affect the spread of HV among host animals also requires further study. From this, weather–host–human associations can be studied in relation to the activity patterns of host animals.

The following limitations of this study should be mentioned. First, the epidemiological data record requirement was standardized in 2005 when the Chinese government established the Notifiable Disease Reporting System (NDRS). Certain reports before 2005 would inevitably contain inaccurate information due to a lack of standardized procedures. Second, solely morphological identification was used to identify the host animal species and historical data are hard to confirm, which could lead to inaccuracies. Third, the lack of serological data on the patients made it challenging to ascertain the interactions of the specific hantavirus species between host animals and humans.

## 5. Conclusions

Our study found that the incidence of HFRS in Zhejiang Province has been on a steady downward trend since 1990. The current intensity of the epidemic was characterized by a high degree of dissemination throughout the region, with annual cases occurring centrally in the summer and winter seasons. It is noteworthy that the incidence population of HFRS tends to be aging. For host animal surveillance, we reviewed the dynamics of host animal population changes and HV infections in various populations in Zhejiang Province over a 32-year period. Firstly, the overall number of animals active in the surveillance area has not decreased significantly, and especially the populations of striped field mice remain at a high level. Secondly, the prevalence of HV in the host animals remains relatively high, with frequent positive results in all types of animals. The lack of a decrease in the HV carriage rate of the striped field mice coupled with their high numbers suggests that the striped field mice have been at risk of causing human epidemics of HFRS. Considering the fact that striped field mice are mainly active in the wild, this implies that people working in agriculture are at a high risk of infection. Moreover, the emergence of HV infection in new host species warranted further surveillance of the HV genotypes. Based on the epidemiological characteristics of HFRS, maintaining a low population density of the host animals in rural areas is a vital measure for preventing HFRS.

## Figures and Tables

**Figure 1 viruses-16-00145-f001:**
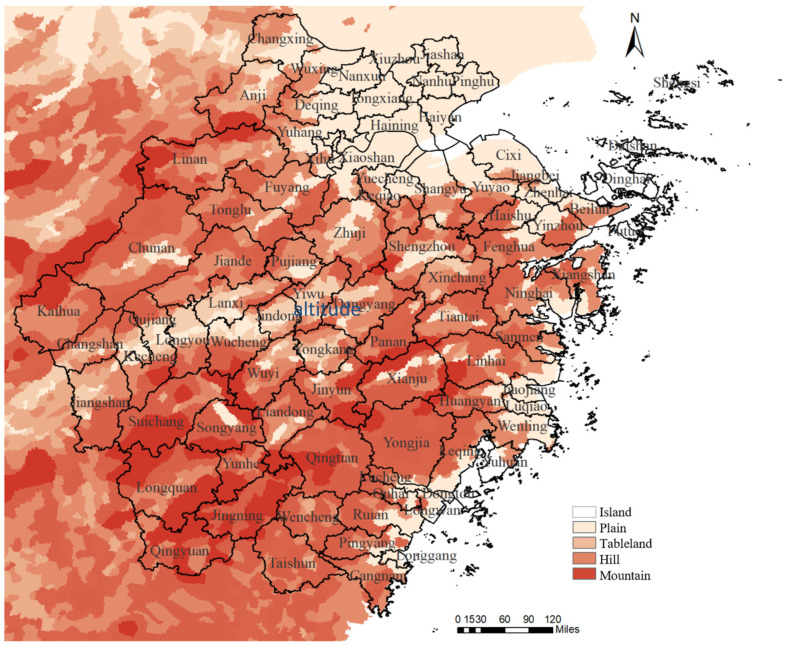
The topographic map of Zhejiang Province.

**Figure 2 viruses-16-00145-f002:**
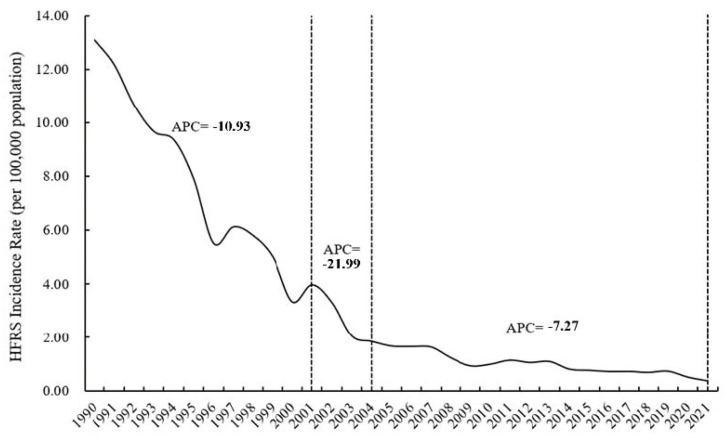
The incidence rate of HFRS in Zhejiang Province, 1990–2021.

**Figure 3 viruses-16-00145-f003:**
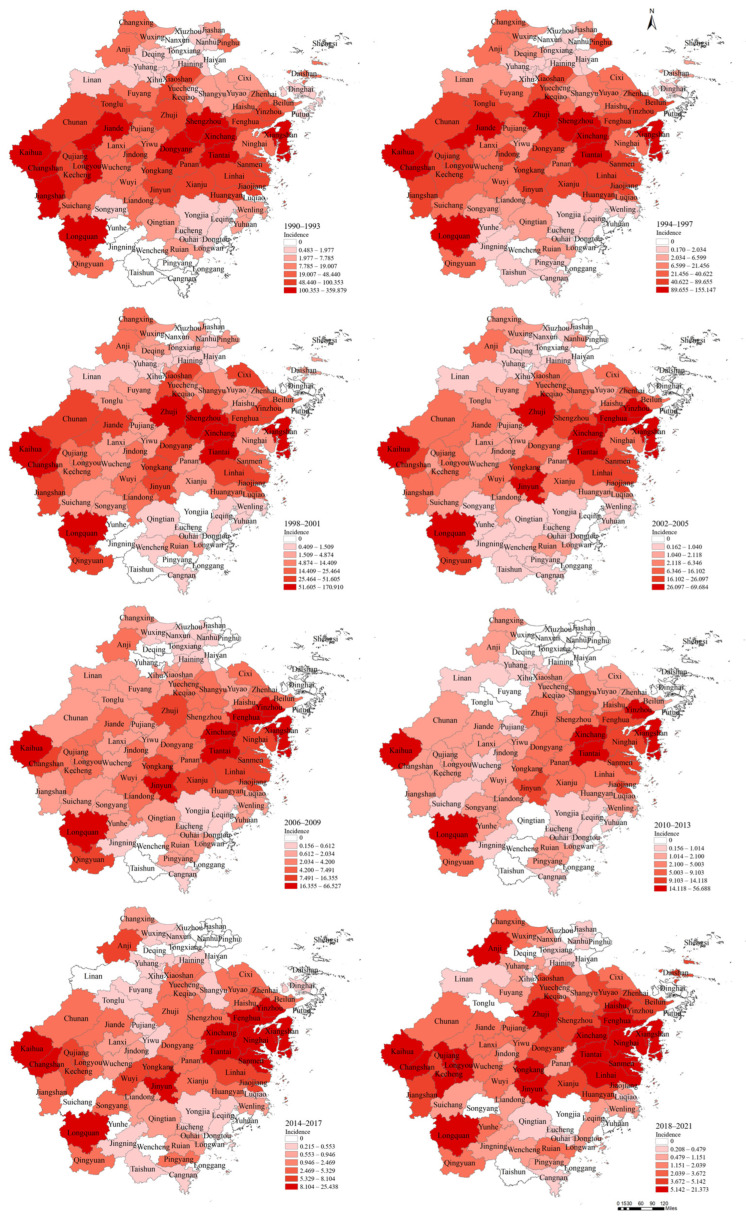
The HFRS incidence rate of each county from 1990 to 2021.

**Figure 4 viruses-16-00145-f004:**
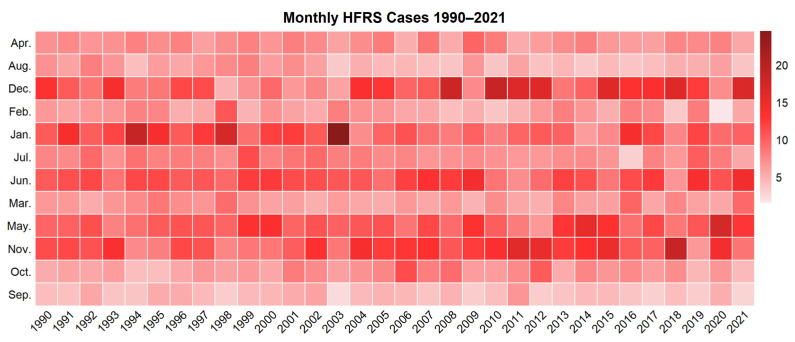
Heatmap of monthly HFRS cases in Zhejiang Province, 1990–2021.

**Figure 5 viruses-16-00145-f005:**
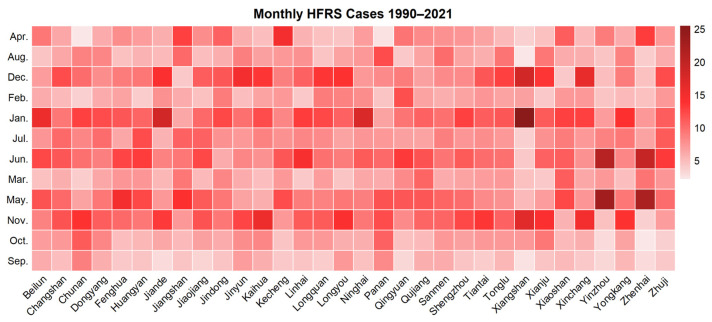
Heatmap of monthly HFRS cases in high-incidence areas, 1990–2021.

**Figure 6 viruses-16-00145-f006:**
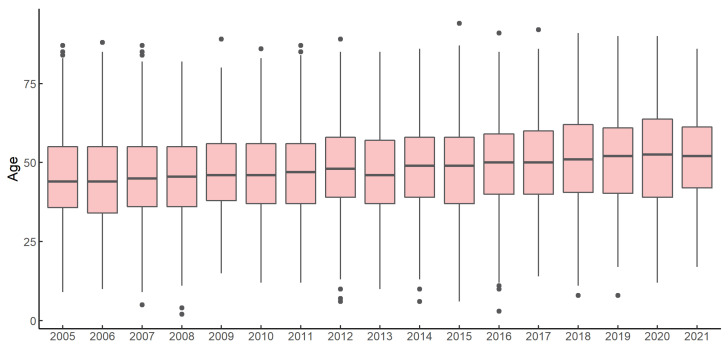
The age distribution of HFRS cases, 2005–2021.

**Figure 7 viruses-16-00145-f007:**
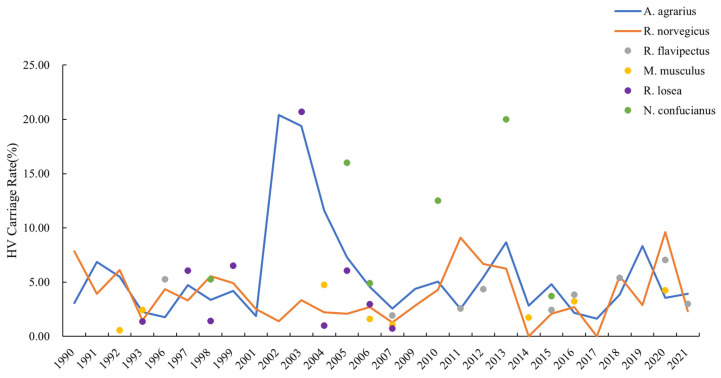
Hantavirus carriage rate in rodents.

**Table 1 viruses-16-00145-t001:** Relative density and HV carriage rate, 1990–2021.

Year	Trap	Captured	Relative Density (%)	Lung	HV Antigen Positive	HV Carriage Rate (%)
1990	8517	839	9.85	757	57	7.52
1991	4766	349	7.32	-	-	-
1992	15,251	1602	10.50	1985	94	4.73
1993	10,320	1063	10.30	705	18	2.55
1994	2200	217	9.86	-	-	-
1995	6800	694	10.20	-	-	-
1996	7048	403	5.72	403	12	2.97
1997	13,139	535	4.07	521	19	3.64
1998	11,020	788	7.15	801	31	3.87
1999	16,162	853	5.27	771	19	2.46
2000	-	-	-	-	-	-
2001	11,519	436	3.78	240	7	2.91
2002	10,059	412	4.09	292	42	14.38
2003	9176	312	3.40	420	66	15.71
2004	11,638	593	5.09	736	49	6.65
2005	13,452	639	4.75	1048	58	5.53
2006	18,474	704	3.81	1074	45	4.18
2007	20,421	649	3.17	1194	17	1.42
2008	20,709	831	4.01	-	-	-
2009	13,294	436	3.27	522	13	2.49
2010	15,116	458	3.02	570	17	2.98
2011	15,284	593	3.87	726	19	2.61
2012	13,296	545	4.09	667	27	4.04
2013	13,785	546	3.96	669	42	6.27
2014	11,694	535	4.57	621	12	1.93
2015	11,995	547	4.56	629	21	3.33
2016	11,271	531	4.71	622	15	2.41
2017	10,053	526	5.23	607	7	1.15
2018	11,199	608	5.42	608	21	3.45
2019	7358	610	8.29	610	35	5.73
2020	10,628	618	5.81	618	25	4.04
2021	11,921	660	5.53	660	15	2.27
Total	367,565	19,132	5.20	19,076	803	4.20

Data from 1991, 1994, 1995, 2000, and 2008 were lost.

**Table 2 viruses-16-00145-t002:** Major host animal species captured in Zhejiang Province.

Year	Rodent							Shrew		Others ^3^
	*A. agrarius*	*R. norvegicus*	*R. losea*	*M. musculus*	*R. flavipectus*	*N. confucianus*	Others ^1^	*S. murinus*	Others ^2^	
1990	456	497	5	70	12	15	98	60	23	0
1991	526	433	63	103	10	20	1	11	35	3
1992	980	605	153	172	57	7	35	18	15	1
1993	443	337	146	41	63	2	16	6	9	0
1996	170	184	10	10	19	1	5	3	1	0
1997	275	121	33	8	11	22	3	3	6	0
1998	387	288	70	30	46	19	45	0	6	0
1999	263	305	46	146	21	6	44	3	14	0
2001	266	80	25	39	10	4	45	2	11	0
2002	196	144	24	30	3	0	10	1	2	11
2003	299	60	29	12	52	6	40	30	3	0
2004	388	90	100	21	117	17	16	18	3	0
2005	385	289	99	111	71	75	57	0	3	6
2006	372	297	168	62	102	41	45	63	7	17
2007	388	313	137	93	104	26	33	57	27	0
2008	433	217	215	47	94	10	90	11	4	0
2009	229	106	91	17	45	7	11	16	0	0
2010	258	70	70	33	27	8	60	35	0	0
2011	315	110	115	98	39	4	33	11	0	1
2012	312	135	83	58	23	7	37	12	0	0
2013	334	128	63	60	30	14	37	5	0	7
2014	353	91	32	57	35	18	15	18	0	2
2015	333	96	20	39	41	54	32	14	0	0
2016	323	111	33	31	52	20	30	22	0	0
2017	370	62	23	46	55	19	15	10	0	7
2018	443	36	12	5	37	44	9	18	4	0
2019	384	69	15	41	29	33	10	25	4	0
2020	310	52	14	47	71	62	37	18	0	7
2021	280	86	34	47	67	72	62	9	3	0
Total	10,471	5412	1928	1574	1343	633	971	499	180	62

^1^ *Microtus fortis*, *Rattus nitidus*, *Niviventer fulvescens*, *Eothenomys melanogaster*, *Cricetulus barabensis*, *Niviventer coninga*, *Berylmys bowersi*, *Apodemus peninsulae*, *Micromys minutus*, *Rattus rattus*, and *Apodemus draco*. ^2^ *Crocidura attenuate* and *Crocidura suaveolens*. ^3^ *Mustela sibirica* and unidentified species.

## Data Availability

The research data are available upon reasonable request.

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
