# Peer review of "Epidemiology of Hemorrhagic Fever with Renal Syndrome and Host Surveillance in Zhejiang Province, China, 1990–2021"

_viruses, 2024, doi:10.3390/v16010145_

Round 1
Reviewer 1 Report
Comments and Suggestions for Authors
Epidemiology of hemorrhagic fever with renal syndrome, and host surveillance in Zhejiang Province, China, 1990-2021.
Fan Su, Ying Liu, Feng Ling, Rong Zhang, Zhen Wang, Jimin Sun
Scientific Content Comments:
1) What is the “Patch Size” mentioned in section 2.3 on line #90. Please describe.
2) Although not necessary for this paper, the authors may want to consider analysis of fecal pellets from trapped rodents for HV, in addition to lung immunofluorescence assays, for HV detection given that hantaviruses are transmitted via rodent feces.
3) In the discussion, line 223-224, the authors compare Zhejiang province with Hunan, Jiangxi and Fujian provinces and continue to show results for other provinces in China. It would be helpful to include the incidence rate graphs for Hunan, Jiangxi and Fujian provinces in Figure 1 since Zhejiang province is compared with other areas in China throughout the manuscript.
4) In Figure 2, the date ranges, ie 1990-1991, 1994-1997, etc) need to be larger. The small type size gets lost in the figure. Additionally, it would be helpful to label each map as A, B, C, etc and then reference the date ranges in the figure legend for Figure 2.
5) Figure 6 the authors shows the carriage rate of HV in rodents. The percent of A. agarius infected with HV spikes between 2001 – 2006. Do the authors know what caused this spike in carriage rate? This should be mentioned in the discussion.
6) On line 147 of section 3.1.3, the authors state that 81.28% of the HFRS cases occur in people age 20-59. 1) Is this the distribution in Zhejiang province or for all of China? 2) What is the distribution of occupations within this age group or is this what is referred to when the authors say, in lines 151-152 that 76.94% of the cases were farmers and 9.35% factory workers. Please clarify.
7) When rodent hosts are mentioned, the common names are used in the text while the scientific names (genius/species) names are listed in Table 2. The first time each of these rodents is mentioned in the text, the common name AND the scientific name (genius/species) name should be given. Scientific names should be in italics. Thereafter, the common name is sufficient. Please correct throughout the manuscript where appropriate.
8) The authors mention that meteorological events vary little between the regions of Zhejiang Province but that there is an abundance of different topographies (hills, plains, islands) in the province (lines 318-330). Do the host animal species, the HV carriage rate within those species and the incidence of human disease differ across those topographies? Showing the relationship between these three components is necessary to understanding the epidemiology of hantavirus disease and should be included (ie Figure 7). This could be accomplished using three heat maps, similar to Figure 1, of Zhejiang Province along with a topographical map of the province to show the relative distributions of 1) host animal species, 2) the HV carriage rate within those species and 3) the incidence of human disease in the province. The authors do partially discuss this in lines 223-241 but a figure that the authors could refer to is important, especially to readers who are unfamiliar with Zhejiang Province.
9) Line #210-211: What is the new “crown” pandemic and what are the dates of this pandemic?
10) Line #278: The authors mention that HV is highly prevalent in animals in China. Would the authors please comment on 1) what is known about how HV carriage rates in Zhejiang Province compare to other provinces in China, and 2) how do HV carriage rates compare to carriage rates for other rodent borne viruses in Zhejiang Province or China in general. This could be accomplished in one or two sentences. Furthermore, HVs that cause HFRS are also found in other Pacific Rim countries (ie the North and South Korea, Vietnam, Thailand) as well as parts of northern Europe. Is it known how the HV carriage rates in rodents in Zhejiang Province or China in general, compare with the HV carriage rates in these other countries? Again, one or two sentences discussing this should be sufficient.

Comments on the Quality of English LanguageEpidemiology of hemorrhagic fever with renal syndrome, and host surveillance in Zhejiang Province, China, 1990-2021.
Fan Su, Ying Liu, Feng Ling, Rong Zhang, Zhen Wang, Jimin Sun
Grammatical Comments:
(requested changes in underlined text)
Line #26: hantaviruses
Line #26: “rodent-borne” infectious disease
Line #32: …the most endemic country for HFRS [3]. Between 2004 and…
Line #38-39. The sentence that begins, “A new source…” on line 38 and finishes on line 39 is essentially the same as the prior sentence. Select one of these sentences.
Line #40: Every HV genotype that is now known to exist can causer HFRS… Is this true of HCPS viruses also, such as ANDV and SNV, or do you mean to say,, “Every Old World HV genotype…”? I would argue that this statement is true primarily for Old World HVs even though New World HVs that cause HCPS can infect the kidneys.
Line #57: “…affect HFRS outbreaks [14]. Moreover, new species…”
Line #59: List the two new insectivorous shrews here.
Line #61: “…animal monitoring in 15 other countries…”
Line #88: “Data on rodent hosts within…”
Line #212: “…measures for the COVID-19 pandemic also contributed…”
Line #214: “…marked increase in the age of the incidence…”
Author Response
Dear Editors and Reviewers:
Thank you for your comments on our manuscript entitled “Epidemiology of hemorrhagic fever with renal syndrome, and host surveillance in Zhejiang Province, China, 1990-2021” (Manuscript ID: viruses-2766854). Those comments are all valuable and very helpful for revising and improving our paper. We have studied comments carefully and have made correction which we hope meet with approval. All the contributing authors have reviewed the revision and concur in the revised submission.
Reviewer 1
- What is the “Patch Size” mentioned in section 2.3 on line #90. Please describe.
Response: Thank you for pointing this out. We placed the traps within a 500-meter radius of the selected sites, so each patch size was about 7,850 square meters. The corresponding modifications are as follows (Lines 108-109):
“More than 200 traps per patch (about 7,850 m2) were placed in the field for three consecutive nights.”
- Although not necessary for this paper, the authors may want to consider analysis of fecal pellets from trapped rodents for HV, in addition to lung immunofluorescence assays, for HV detection given that hantaviruses are transmitted via rodent feces.
Response: Thank you for your very interesting comment. In the future, we will try to detect fecal pellets from trapped rodents for HV.
- In the discussion, line 223-224, the authors compare Zhejiang province with Hunan, Jiangxi and Fujian provinces and continue to show results for other provinces in China. It would be helpful to include the incidence rate graphs for Hunan, Jiangxi and Fujian provinces in Figure 1 since Zhejiang province is compared with other areas in China throughout the manuscript.
Response: Thank you for the comment. Unfortunately, due to the lack of raw data from other provinces, we were not able to graphically represent the incidence rates outside of Zhejiang Province. All references to other regions in this paper are taken from the published literature, and the corresponding articles are listed in the references.
- In Figure 2, the date ranges, ie 1990-1991, 1994-1997, etc) need to be larger. The small type size gets lost in the figure. Additionally, it would be helpful to label each map as A, B, C, etc and then reference the date ranges in the figure legend for Figure 2.
Response: Thanks for the comment and we revised Figure 2.
- Figure 6 the authors shows the carriage rate of HV in rodents. The percent of A. agarius infected with HV spikes between 2001-2006. Do the authors know what caused this spike in carriage rate? This should be mentioned in the discussion.
Response: This issue also haunts us. A study suggested that climatic factors, such as temperature and precipitation, may influence the population dynamics and virus transmission of rodents. Another study noted that the sampling methods and locations may affect the estimation of HV carriage rate among rodents. Because our study did not include meteorological factors and the data sources are long dated, we believe that both could have contributed to the above results. We added some sentences in the discussion section (Lines 321-325).
“ Tagliapietra et al1 suggested that climatic factors, such as temperature and precipitation, may influence the population dynamics and virus transmission of rodents. The spike in HV carriage rate among rodents may also be partly due to the variation in sampling design and quality, rather than the actual change in virus prevalence. ”
- On line 147 of section 3.1.3, the authors state that 81.28% of the HFRS cases occur in people age 20-59. 1) Is this the distribution in Zhejiang province or for all of China? 2) What is the distribution of occupations within this age group or is this what is referred to when the authors say, in lines 151-152 that 76.94% of the cases were farmers and 9.35% factory workers. Please clarify.
Response: Thank you for your valuable comment. “81.28% of the HFRS cases occur in people age 20-59.” the distribution is in Zhejiang Province. And Of those aged 20-59, 67.83% were farmers and 15.75% factory workers. The corresponding modifications are as follows (Lines 170-172):
“Overrall, farmers accounted for 76.94% of the cases, with factory workers accounting for 9.35%. Of those aged 20-59, 67.83% were farmers and 15.75% factory workers.”
- When rodent hosts are mentioned, the common names are used in the text while the scientific names (genius/species) names are listed in Table 2. The first time each of these rodents is mentioned in the text, the common name AND the scientific name (genius/species) name should be given. Scientific names should be in italics. Thereafter, the common name is sufficient. Please correct throughout the manuscript where appropriate.
Response: Thank you for your reminding. We have made modifications throughout the manuscript.
- The authors mention that meteorological events vary little between the regions of Zhejiang Province but that there is an abundance of different topographies (hills, plains, islands) in the province (lines 318-330). Do the host animal species, the HV carriage rate within those species and the incidence of human disease differ across those topographies? Showing the relationship between these three components is necessary to understanding the epidemiology of hantavirus disease and should be included (ie Figure 7). This could be accomplished using three heat maps, similar to Figure 1, of Zhejiang Province along with a topographical map of the province to show the relative distributions of 1) host animal species, 2) the HV carriage rate within those species and 3) the incidence of human disease in the province. The authors do partially discuss this in lines 223-241 but a figure that the authors could refer to is important, especially to readers who are unfamiliar with Zhejiang Province.
Response: Thank you for your suggestion. Unfortunately, the geographic information included in the historically recorded case data and animal surveillance data is the administrative division of Zhejiang Province. The smallest geographic unit involved in this study was the county, but multiple topographies also exist in a county. As a result, we will carry out investigations in future studies using topographic features as the basis for area categorization. As a reference, we have added a topographic map of Zhejiang Province in 2.1 Study Sites.
- Line #210-211: What is the new “crown” pandemic and what are the dates of this pandemic?
Response: We are very sorry for our inappropriate expression of the sentence. The modifications are as follows (Lines 231-235)
“During the COVID-19 pandemic, the incidence rate in Zhejiang Province dropped to 0.49 per 100,000 population in 2020, and the preventive and control measures for the COVID-19 pandemic also contributed to a sudden decrease in the incidence of various zoonotic diseases.”
- Line #278: The authors mention that HV is highly prevalent in animals in China. Would the authors please comment on 1) what is known about how HV carriage rates in Zhejiang Province compare to other provinces in China, and 2) how do HV carriage rates compare to carriage rates for other rodent borne viruses in Zhejiang Province or China in general. This could be accomplished in one or two sentences. Furthermore, HVs that cause HFRS are also found in other Pacific Rim countries (ie the North and South Korea, Vietnam, Thailand) as well as parts of northern Europe. Is it known how the HV carriage rates in rodents in Zhejiang Province or China in general, compare with the HV carriage rates in these other countries? Again, one or two sentences discussing this should be sufficient.
Response: Based on the results of previous studies and the present study, we concluded that 1) HV carriage rates in Zhejiang Province are relatively high compared to other provinces in China. 2) HV carriage rates are higher than carriage rates for other rodent borne viruses in Zhejiang Province or China in general. 3) HV carriage rates in rodents in China in general are higher than those in other countries.These are expressed in the text as follows (Lines 297-298, 302-303)
“HV carriage rates are higher than carriage rates for other rodent-borne viruses. And the rates in China are generally higher than in other countries.”
“ HV carriage rates in Zhejiang Province are relatively high compared to other provinces in China. “
- Line #26: hantaviruses
Line #26: “rodent-borne” infectious disease
Line #32: …the most endemic country for HFRS [3]. Between 2004 and…
Line #38-39. The sentence that begins, “A new source…” on line 38 and finishes on line 39 is essentially the same as the prior sentence. Select one of these sentences.
Line #40: Every HV genotype that is now known to exist can causer HFRS… Is this true of HCPS viruses also, such as ANDV and SNV, or do you mean to say,, “Every Old World HV genotype…”? I would argue that this statement is true primarily for Old World HVs even though New World HVs that cause HCPS can infect the kidneys.
Line #57: “…affect HFRS outbreaks [14]. Moreover, new species…”
Line #59: List the two new insectivorous shrews here.
Line #61: “…animal monitoring in 15 other countries…”
Line #88: “Data on rodent hosts within…”
Line #212: “…measures for the COVID-19 pandemic also contributed…”
Line #214: “…marked increase in the age of the incidence…”
Response: We revised this grammatical comments. Thank you again for your valuable comments.
Reviewer 2
- r 25 Hemorrhagic fever with renal syndrome (HFRS),a rodent infectious disease
Comment: multiple errors in the first sentences. HFRS is a syndrome not a rodent infectious disease caused by old world hantaviruses. Please rewrite the introduction up to r 32. And take the reader by hand, explain HFRS explain Hanta use some of the big reviews like https://www.thelancet.com/journals/laninf/article/PIIS1473-3099(23)00128-7/fulltext and https://pubmed.ncbi.nlm.nih.gov/22670688/
Response: Thank you for your valuable comment. We made the following modifications (Line30-36):
“Hemorrhagic fever with renal syndrome (HFRS),a rodent-borne disease caused by hantaviruses (HV) of the family Bunyaviridae, is characterized by fever, hemorrhage, kidney damage and hypotension. According to previous studies, over 90% of all cases of HFRS in the world have occurred in China in recent decades. Despite the recent reduction in human morbidity and mortality from HFRS as a result of improved prevention and treatment strategies, HFRS remains a significant public health challenge in China. To date, China remains the most endemic country for HFRS.”
- 2 data collection
comment: as a reviewer I want more details on when a patient is scored as HFRS was it by clinicians choice, positieve PCR, serology or just when HFRS is mentioned in patient files? this matters a lot, could it be that clinicians forgot about HFRS and didn't test anymore? all your conclusions also depend on the knowledge, attitude and test behavior of your countries clinicians
Response: We apologize for not clarifying the issue you mentioned. The corresponding modifications are as follows (Lines 88-97):
“According to the health industry standard of the People’s Republic of China for diagnostic criteria of HFRS, HFRS cases were classified as suspected cases, clinically diagnosed cases and confirmed cases. (1) A patient with epidemiological history and clinical manifestations with fever or gastrointestinal symptoms was defined as a suspected case; (2) A suspected case with hypotension, renal function impairment, increased peripheral blood cell counts and thrombocytopenia, and a positive urine protein was defined as a clinically diagnosed case; (3) A clinically diagnosed case with one or more of the following criteria: the serum specific IgM antibody is positive; the specific IgG antibody is 4 times higher than that in the acute phase; Hantavirus RNA is positive or Hantavirus is isolated was defined as a confirmed case.”
- discussion should be improved to long now and no focus.
Response: Thank you for your suggestion. We revised the discussion section.
- Also do you believe there might be a role for soil mineral changes like postulated in https://pubmed.ncbi.nlm.nih.gov/25609306/
Response: This is very enlightening. We do not currently have access to the distribution of selenium in the various regions of Zhejiang Province, and will attempt to explore the link between the two after subsequent investigations.
- delete the abstract and start over. This one does not invite the reader to read your well prepared manuscript and lacks quality which you do show in your manuscript
Response:Thank you for your comments.We made the following modifications (Abstract):
“Hemorrhagic fever with renal syndrome (HFRS) is caused by hantaviruses (HV) and is endemic in Zhejiang Province, China. In this study, we aimed to explore the changing epidemiology of HFRS cases and dynamics of hantavirus host in Zhejiang Province. The joinpoint regression for analyzing long-term trends in the incidence of HFRS. Comparison of animal density at different stages using the Mann-Whitney Test. A comparison of HV carriage rates between stages and species was performed using the chi-square test. The incidence of HFRS shows a continuous downward trend. Cases are widely distributed in all counties of Zhejiang Province except Shengsi County. There was a high incidence belt from west to east, with low incidence in the south and north. The HFRS epidemic showed two seasonal peaks in Zhejiang Province, which were winter and summer. It showed a marked increase in the age of the incidence population. A total of 23,073 minibeasts from 21 species were captured. Positive results were detected in lung tissues of 14 rodent species and 1 shrew species. 80% of the positive results were from the striped field mice and the brown rats. No difference in HV carriage rates between striped field mice and brown rats (χ2=0.258, p=0.611).”
Thank you again for your valuable comments.
We have tried our best to improve the manuscript and made some changes in the manuscript. We appreciate for Editors/Reviewers’ warm work earnestly, and hope that the correction will meet with approval.
Once again, thank you very much for your comments and suggestions.
Yours sincerely,
Jimin Sun

Reviewer 2 Report
Comments and Suggestions for Authors
Su et al describe the changing epidemiology of HFRS in Zhejiang Province in China. This extensive paper can be of interest for the readers of viruses. Especially due to the good standard of tests and visualization of the results. However, the text needs to be improved.
Also the occurrence of COVID-19 and the preventive measurements create a unique situation where multiple hypothesis can be tested. The authors now only use line 212 to shortly mention covid. But this could be the major strength of the paper to add analysis comparing the specific covid period with periods before and after covid. To be honest although of high quality, the authors remain a bit on the safe side and I would like to challenge them to come up with bit more exciting analysis and observations.
minor changes:
r 25 Hemorrhagic fever with renal syndrome (HFRS),a rodent infectious disease
Comment: multiple errors in the first sentences. HFRS is a syndrome not a rodent infectious disease. caused by old world hantaviruses. Please rewrite the introduction up to r 32. And take the reader by hand, explain HFRS explain Hanta use some of the big reviews like https://www.thelancet.com/journals/laninf/article/PIIS1473-3099(23)00128-7/fulltext
and https://pubmed.ncbi.nlm.nih.gov/22670688/
2.2 data collection
comment: as a reviewer I want more details on when a patient is scored as HFRS was it by clinicians choice, positieve PCR, serology or just when HFRS is mentioned in patient files? this matters a lot, could it be that clinicians forgot about HFRS and didn't test anymore? all your conclusions also depend on the knowledge, attitude and test behavior of your countries clinicians
discussion should be improved
to long now and no focus.
Also do you believe there might be a role for soil mineral changes like postulated in https://pubmed.ncbi.nlm.nih.gov/25609306/
Comments on the Quality of English Language
delete the abstract and start over. This one does not invite the reader to read your well prepared manuscript and lacks quality which you do show in your manuscript
Author Response

(The authors gave the same response as above.)
